# What defines a photosynthetic microbial mat in western Antarctica?

**Ricardo A. Mercado-Juárez[1,2], Patricia M. Valdespino-Castillo[3,4], Martín Merino Ibarra[5], Silvia Batista[6], Walter Mac Cormack[7,8], Lucas Ruberto[7,8], Edward J. Carpenter[9], Douglas G. Capone[10], Luisa I. Falcón[2]***

1 Posgrado en Ciencias Biológicas, Universidad Nacional Autónoma de México, Unidad de Posgrado, Coyoacán, México, 2 Laboratorio de Ecología Bacteriana, Instituto de Ecología, Unidad Mérida, UNAM, Ucú, México, 3 Molecular Biophysics and Integrated Bioimaging Division, Lawrence Berkeley National Laboratory, Berkeley, California, United States of America, 4 Escuela Nacional de Ciencias de la Tierra, UNAM, Coyoacán, México, 5 Unidad Académica de Biodiversidad Acuática, Instituto de Ciencias del Mar y Limnología, UNAM, Coyoacán, México, 6 Instituto de Investigaciones Biológicas Clemente Estable, Montevideo, Uruguay, 7 Instituto Antártico Argentino, Buenos Aires, Argentina, 8 Instituto NANOBIOTEC UBA-CONICET, Facultad de Farmacia y Bioquímica, Universidad de Buenos Aires, Argentina, 9 Estuary and Ocean Science Center, San Francisco State University, Tiburon, California, United States of America, 10 Department of Biological Sciences, Marine and Environmental Biology Section, University of Southern California, Los Angeles, California, United States of America

* falcon@ecologia.unam.mx

## Abstract

Antarctic microbial mats, with their significant biodiversity and key role in biogeochemical cycling, were the focus of our study. We employed a metagenomic approach to analyze 14 microbial mats from meltwater streams of western Antarctica, covering the Maritime, Peninsula, and Dry Valleys regions. Our findings revealed that the taxonomic compositional level of the microbial mat communities is characterized by similar bacterial groups, with diatoms being the main distinguishing factor between the rapidly warming Maritime Antarctica and the other mats. Bacteria were found to be the predominant component of all microbial mats (>90%), followed by Eukarya (>3%), Archaea (<1%), and Viruses (<0.1%). The average abundance of the main phyla composing Antarctic microbial mats included Bacteroidota (35%), Pseudomonadota (29%), Cyanobacteriota (19%), Verrucomicrobiota (3%), Bacillariophyta (2%), Planctomycetota (2%), Acidobacteriota (2%), Actinomycetota (2%), Bacillota (1%), and Chloroflexota (1%). We also identified some microeukaryotes that could play essential roles in the functioning of Antarctic microbial mats. Notably, all mats were found in sites with varied environmental characteristics, showed N-limitation, and shared functional patterns.

## Introduction

Freshwater ecosystems in Antarctica are considered biodiversity hotspots. However, anticipated warming is expected to alter environmental conditions, which could affect this biodiversity. Antarctica and the Southern Ocean are particularly sensitive to climatic influences, and the physical changes resulting from climate change are likely to impact life both in Antarctica and globally [1]. It is important to note that Antarctica is undergoing rapid environmental

**Data availability statement:** All sequences are available in EMBL-EBI data repositories. All sequencing data are available at the European Nucleotide Archive (ENA) under the accession number PRJEB14287 and PRJEB12762.

**Funding:** Financing was granted by Agencia Mexicana de Cooperación Internacional para el Desarro-llo/Agencia Uruguaya de Cooperación Internacional (AMEXCID/AUCI -URU001) to SB and LIF, AMEXCID/Dirección Nacional del Antártico (AMEXCID/DNA -ARG001) to WMcM and LIF. DGC and EJC thank the Division of Polar Programs, National Sciences Foundation (NSF) for sustained support. The funders had no role in study design, data collection and analysis, decision to publish, or preparation of the manuscript.

**Competing interests:** The authors have declared that no competing interests exist.

change, which may alter ecological gradients and affect the continent's biodiversity [2,3]. Simulated effects of similar physical changes have already been studied, such as increased availability of liquid water. In one experiment, stream water was diverted to an adjacent area of arid soil, and changes in microbial composition and activity were monitored over seven weeks using molecular and biochemical methods [4], The effects of nutrient amendments have also been tested, revealing that microbial mats exhibit functional and structural responses to nutrient inputs [5]. Furthermore, environmental variables are the best predictors of microbial community composition [6,7]. In this context, biogeochemical cycles may be altered, leading to rapid environmental changes impacting Antarctic biota. Therefore, inventories of freshwater ecosystem diversity in Antarctica are essential for understanding future climate change impacts.

Microbial mats are complex communities of microorganisms found in various extreme environments [8]. In Antarctica, these mats are typical in ice-free regions, particularly in environments where water flows due to snowmelt during the austral summer [9]. In freshwater environments, microbial life dominates, ranging from biofilms to microbial mats [10]; Microbial mats can thrive under extreme and variable conditions by incorporating biomass and nutrients, alleviating the nutrient-poor conditions of polar oligotrophic landscapes [1,11–13]. These mats comprise bacteria, microalgae, fungi, and microeukaryotes, with prokaryotes predominating in diversity [9,10,14,15]; Common microeukaryotes in microbial mats include phototrophs, diverse heterotrophic protists, fungi, tardigrades, nematodes, and rotifers [16–18]. Cyanobacteria and microalgae are the leading primary producers [19–22], while protists are typically consumers within the mats [23, 24]. Compared to microeukaryotes, prokaryotes dominate terrestrial habitats on the Antarctic continent [4,25–28]. Although several studies on Antarctic microbial mats exist, most focus on bacteria [7,29–31].

Microeukaryotes, though not highly diverse within microbial mats, perform critical ecological functions. Diatoms can be used as environmental indicators in Antarctic environments due to their close correlation with conductivity [32] and as biogeographic predictors due to their limited dispersal and high level of endemism at higher latitudes [33]. Organisms such as nematodes, tardigrades, and rotifers play vital roles in trophic levels, mainly as consumers, while rotifers, in particular, contribute to biomass production [34]. Fungi act as decomposers and can modify the functioning of microbial mats. Increased temperatures promote fungal growth, which may alter the structure and resilience of the mats, reduce nitrogen levels, and generally impact their composition [35]. Studies focused on microeukaryotes often use non-culture-based approaches to investigate sediment communities in marine environments in Western Antarctica [36], conduct paleobiological reconstructions of microbial mats at local sites like McMurdo Ice Shelf [37], and identify eukaryotes in cyanobacterial mat communities in both Antarctica and the extreme High Arctic [17]. However, no studies had analyzed Western Antarctica microbial mats at a continental scale using shotgun sequencing technology, which we coupled in this study with a physicochemical characterization, to describe and compare sites including Maritime Antarctica, the Antarctic Peninsula, and the McMurdo Dry Valleys.

## Materials and methods

### Study sites

Microbial mats were collected in three regions of western Antarctica (60 to 80º S) including King George Island (25 de mayo), which is the largest of the South Shetland Islands in Maritime Antarctica (MA), Antarctic Peninsula (AP) (also part of MA and differentiated as Peninsula for clarity), and McMurdo Dry Valleys (DV) in continental Antarctica (Fig.1, Table 1) [38, 39].

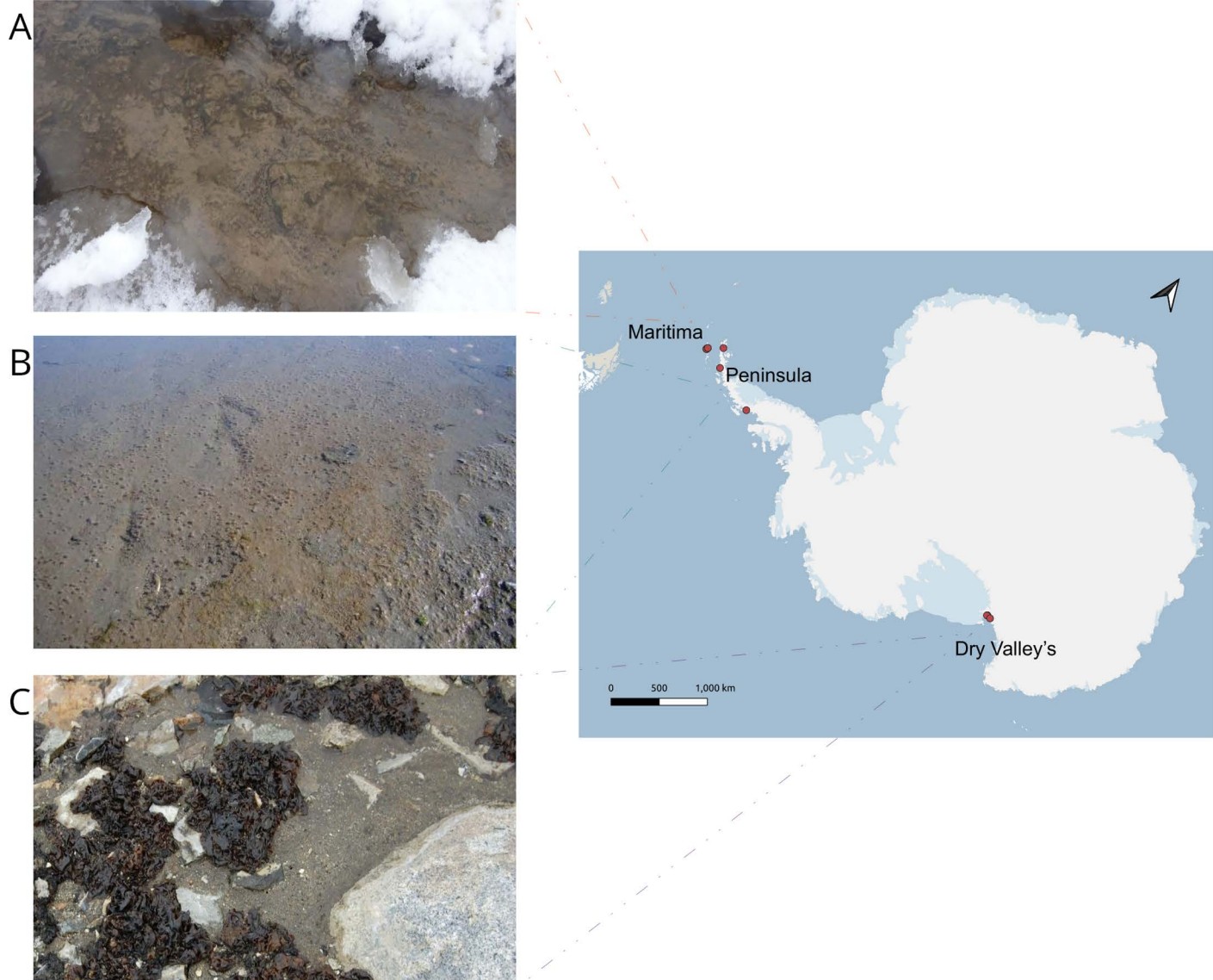

**Fig 1. Pictures and location of the microbial mats collected.** *In situ* images of the sampled microbial mats. (A) Fildes in Maritime Antarctica, (B) Esperanza in Antarctic Peninsula, and (C) Garwood in McMurdo Dry Valleys. Location of sampling sites in western Antarctica: Maritime, Antarctic Peninsula, and McMurdo Dry Valleys. The red dots represent the geographic sites where microbial mats and water were collected. Map was obtained from the Quantarctica package (http://quantarctica.npolar.no/) and plotted in QGIS v3.28.10 (http://www.qgis.org).

Four of the six MA sampling sites were distributed in the Fildes Peninsula, an ice-free region in summer located in south-western King George Island. Two sites were located near the Carlini Antarctic Base (Argentina), which is located on the south of Potter bay, south west of King George Island. The mean annual temperature in this region is -2 º C, and precipitation reaches c. 800 mm [40]. MA islands experience less extreme daily and seasonal temperature variations than locations within the continent but show more variation in other variables such as precipitation [38].

The Antarctic Peninsula (AP) has an area of c. 420,000 km2 and stretches for 1,300 km from c. 63 to 74°S towards the South Pole. Snow and permanent ice cover c. 97% of the area, and terrestrial communities are restricted to regions of ice-free ground. AP in coastal areas

**Table 1. Nutrient and biomass data for microbial mat samples.**

| | Zone | Lat | Long | SRP | SRSi | $NO_2^-$ | $NO_3^-$ | $NH_4^+$ | DIN | DIN:SRP | $C_{(EA)}$ | $N_{(EA)}$ | $C:N_{(EA)}$ | $N_{(Val)}$ | $P_{(Val)}$ | $N:P_{(Val)}$ |
|---|---|---|---|---|---|---|---|---|---|---|---|---|---|---|---|---|
| | | DD | DD | µM | µM | µM | µM | µM | µM | | % | % | | % | % | |
| Fildes1 | MA | -62.16 | -58.94 | 0.22 | 49.57 | 0.11 | 9.96 | 3.14 | 13.21 | 60.05 | 6.99 | 0.35 | 19.97 | 0.55 | 0.04 | 12.60 |
| Fildes2 | MA | -62.16 | -58.98 | 0.07 | 14.31 | 0.05 | 6.36 | 2.35 | 8.76 | 125.14 | 8.80 | 0.72 | 12.22 | 0.70 | 0.04 | 18.71 |
| Fildes3 | MA | -62.20 | -58.95 | 0.32 | 63.44 | 0.17 | 28.93 | 2.00 | 31.10 | 97.19 | 11.51 | 1.12 | 10.28 | 1.07 | 0.08 | 12.76 |
| Fildes4 | MA | -62.17 | -58.98 | 0.22 | 29.37 | 0.09 | 6.59 | 5.25 | 11.93 | 54.23 | 4.49 | 0.47 | 9.55 | 0.51 | 0.06 | 8.30 |
| Potter5 | MA | -62.24 | -58.67 | 0.50 | 94.36 | 0.22 | 4.17 | 22.60 | 26.99 | 53.98 | 8.89 | 0.94 | 9.46 | 0.54 | 0.28 | 1.98 |
| Potter6 | MA | -62.24 | -58.66 | 0.52 | 18.05 | 0.19 | 4.61 | 68.20 | 73.00 | 140.39 | 38.14 | 8.03 | 4.75 | 2.51 | 0.70 | 3.59 |
| Esperanza | AP | -63.47 | -57.00 | 1.18 | 8.96 | 1.18 | 23.72 | 33.30 | 58.20 | 49.32 | 10.62 | 0.56 | 18.96 | 0.50 | 0.03 | 14.84 |
| Primavera | AP | -64.16 | -60.96 | 26.24 | 12.96 | 1.51 | 57.99 | 354.20 | 413.70 | 15.77 | 50.44 | 6.63 | 7.61 | 2.65 | 0.57 | 4.65 |
| San Martin | AP | -68.13 | -67.10 | 2.13 | 15.73 | 0.37 | 88.53 | 3.87 | 92.77 | 43.55 | 15.91 | 1.61 | 9.88 | 1.45 | 0.24 | 6.04 |
| Garwood10 | DV | -78.02 | 163.92 | 0.12 | 107.90 | 0.22 | 5.45 | 0.71 | 6.38 | 53.61 | 3.28 | 0.44 | 7.46 | 0.38 | 0.08 | 4.73 |
| Garwood11 | DV | -78.02 | 163.92 | 0.25 | 26.64 | 0.02 | 2.67 | 1.65 | 4.34 | 17.48 | 11.93 | 1.49 | 8.01 | 0.99 | 0.08 | 11.91 |
| Garwood12 | DV | -78.02 | 163.90 | 0.08 | 38.81 | 0.05 | 0.01 | 2.41 | 2.46 | 31.15 | 2.73 | 0.40 | 6.83 | 0.24 | 0.04 | 5.52 |
| Garwood13 | DV | -78.03 | 164.10 | 0.65 | 104.26 | 0.03 | 12.40 | 0.21 | 12.64 | 19.42 | 3.15 | 0.32 | 9.84 | 0.30 | 0.06 | 5.36 |
| Taylor | DV | -77.66 | 163.09 | 0.41 | 4.97 | 0.01 | 0.89 | 0.29 | 1.18 | 2.92 | 14.42 | 1.62 | 8.90 | 1.15 | 0.16 | 7.45 |

Nutrients in surrounding water and biomass values of 14 microbial mat samples, including sampling zones and coordinates.

Zone: Martime Antarctica (MA), Antarctic Peninsula (AP), Dry Valley's (DV)

Lat: latitude; Long: longitude; DD: decimal degrees

SRP: Soluble reactive phosphorus; SRSi: Soluble reactive silicon; µM: micromolar

(EA): elemental analysis; (Val): Valderrama method

experience mean air temperatures around 0–3°C during summer [41]. All AP sampling sites were located on rocky shores, which were covered in ice and were close to Argentinian scientific bases (Esperanza, Primavera, and San Martin).

McMurdo Dry Valleys (DV) are the largest ice-free region on the Antarctic continent, with an area of c. 4000 km2, located on the coastal margin between c. 77 to 79° S and c. 160 to 164° E in Victoria Land. The landscape is dominated by arid soils, glaciers, valleys, lakes, and ephemeral streams with melting glaciers in summer. The average temperature is -18°C, with continuous darkness in winter and constant sunlight with high UV radiation in summer. Due to the action of the katabatic winds, there is no snow accumulation or precipitation. DV is considered an extreme polar desert [11]. Sampling was done in Garwood and Taylor valleys as part of the SF Antarctic Organisms and Ecosystems Program. All sites were characterized as cyanobacterial mats and developed under melt water streams.

## Sampling

Antarctic microbial mats analyzed develop in the benthos of meltwater streams and form thick pigmented biofilms dominated by cyanobacteria [42]. Microbial mats were collected as part of a joint sampling effort between Antarctic agencies including the Instituto Antártico Uruguayo (IAU), Instituto Antártico Argentino (IAA), the NSF (USA) and the Mexican Agency for International Cooperation (AMEXCID). All teams conducted parallel sampling, during the austral summer of 2015, following the same strategies and protocols. Each microbial mat was sampled in a 1x1 m² area, with 5x1 g subsamples of 1 cm³ in each corner and center of the mat. All samples were carried out using sterile material and stored aseptically until processed. Fourteen microbial mats (in quintuplets) and their overflowing water were collected including 6 microbial mats in MA, 3 mats in AP and 5 mats in DV. All microbial

mats were collected from ice-melt currents at approximately 0.2 m depth (Fig. 1). Water samples for nutrient analysis were collected in polypropylene acid-washed bottles, in triplicates, and filtered through nitrocellulose membranes (0.45 and 0.22 μm, HA Millipore™). Mat and water samples were collected with sterile material and frozen at -20° C until analysis in the laboratory. Samples were collected under different Antarctic research programs: in Maritime Antarctica, AMEXCID/AUCI; in Peninsula, AMEXCID/IAA and in the Dry Valleys through the NSF polar program. Each sampled mat position was recorded with GPS.

## Nutrients in water and microbial mat biomass

In the laboratory, nutrient determinations of ammonium ($NH_4^+$), nitrate ($NO_3^-$), nitrite ($NO_2^-$), soluble reactive phosphorus (SRP) and soluble reactive silicon (SRSi) were carried out using a Skalar San Plus continuous-flow autoanalyzer following the standard methods adapted by Grasshoff et al. [43] and the circuits indicated by Kirkwood [44]. Dissolved inorganic nitrogen (DIN) was calculated from the sum of ammonium ($NH_4^+$) and nitrogen inorganic forms ($NO_2^-$, $NO_3^-$). The carbon, nitrogen, and phosphorus content (%) in the microbial mat biomass was determined through elemental analysis. Microbial mats were cold-dried and ground in an agate mortar. Approximately 20 mg of material were used to measure elemental carbon and nitrogen with a PerkinElmer 2400 Elemental Analyzer, using five replicates per sample. The nitrogen and phosphorus concentrations were analyzed using 0.05 g of mat sample, following high-temperature persulfate oxidation as described by Valderrama [45].

## DNA extraction and sequencing

Approximately 0.5g of each microbial mat replicate (n=14x5) was used to extract total DNA following the QIAGEN Soil and Power kit instructions. Metagenomic libraries (n==14) were prepared from genomic DNA with a pooled sample for each mat (n = 5), with the Nextera DNA Flex library prep kit (Illumina, San Diego, CA, USA). Fragments of total DNA (1 mg) were inserted into vectors and sequenced with whole genome sequencing technology (HiSeq2x150), at the Yale Keck Center for Genomic Sciences. A mean of 7.82 Gb of paired-end reads data sets was obtained for each metagenome, for a total of 109 Gbps of sequenced DNA. All sequencing data are available at the European Nucleotide Archive (ENA) under the accession number PRJEB14287 and PRJEB12762.

## Bioinformatic processing

Metagenomic raw fastq files were processed by fastp [46] for adapter removal, low-quality filtering, duplicated reads, and trimming of the first 10 bp from PE read [fastp -i raw.1.fq.gz -o qc.1.fq.gz -I raw.2.fq.gz -O qc.2.fq.gz --trim_front1 10 --trim_front2 10 --dedup --dup_calc_accuracy 6]. To assign taxonomic classification to the quality-filtered fastq files, we used Kaiju v1.9.2 [47] with default parameters [kaiju -t nodes.dmp -f kaiju_db_nr_euk.fmi -i qc.1.fq.gz -j qc.2.fq.gz -o kaiju.out] against the NCBI nr 2021-02 database [48]. We used kaiju2table to generate summary tables or taxonomic profile from the kaiju output files at genus level [kaiju2table -t nodes.dmp -n names.dmp -r genus -l superkingdom,phylum,class,order,family,-genus -o kaiju_summary.tsv kaiju.out {kaiju2.out, …}]. SingleM (https://github.com/wwood/singlem) was used to profile the alpha diversity of quality-filtered fastq files from the relative abundances of single marker genes. Specifically, singlem pipeline was used to generate OTUs tables for each metagenome [singlem pipe --forward qc.1.fq.gz --reverse qc.2.fq.gz --otu-table otu.table.tsv]. Particularly, to detect sequences associated with eukaryotes in metagenomic read datasets, we employed Eukdetect [49], and Metaxa2 [50].

The processed and quality control metagenomic data were assembled using MegaHit v1.2.9 [51] with default parameters [megahit -1 qc.1.fq.gz -2 qc.2.fq.gz -o mat.sample]. Contigs greater than 1000 bp were screened in eukaryotic, prokaryotic and unassigned through Kaiju [47] followed by eukaryotic gene prediction using Augustus v3.5.0 [52] with eukaryotic models and prodigal v2.6.3 [53] for prokaryotes; and ORF for unclassified contigs with emboss getorf [54]. Predicted genes for each prediction dataset were then clustered at a 40% identity and 80% coverage threshold using mmseqs2 [55] to avoid redundancy. Functional annotation and orthology assignments of gene clusters were performed using eggNOG Mapper v2 [56] against the eggNOG database [57] with diamond algorithm [58] for the search step; functional genes from the custom database [59] were also annotated. Annotations were transformed into GPM (Gene Per Million) for each hit [(counts of gene X/ total number of paired reads) * 1000000] to enable comparisons. Additionally, rRNA sequences were predicted using Barrnap [60]. These predicted sequences were subsequently annotated, aligned, and classified with ACT Silva [61]. Eukaryote-assigned rRNA sequences were then clustered using cd-hit-est [62], aligned with MAFFT [63], and re-annotated against the nr database using BLASTn [64]. Finally, a phylogenetic reconstruction for *Adineta vaga* was performed with RAxML [65] and visualized in FigTree (http://tree.bio.ed.ac.uk/software/figtree/).

## Statistical analysis and comparison

A principal component analysis (PCA) was performed to evaluate nutrient concentrations in water at 14 sampling sites. Previously, the data were standardized to ensure that the variables had a comparable scale. The PCA was carried out using the princomp function of the R software (version 4.4.1). The resulting biplot, representing both samples and variables in the space of the first two principal components, was visualized using the fviz_pca_biplot from factoextra package in R. The proportions of variance explained by each component were calculated to interpret the importance of each axis in the total variability of the data.

The diversity at genus level was quantified with 14 ribosomal proteins (*rplK, rplN, rplP, rplB, rplC, rplE, rplF, rpsJ, rpsS, rpsB,* S12/S23, S15P, S5, S7) that were filtered from OTU tables to generate Shannon-Wiener and Simpson indexes with diversity from package vegan [66]. Taxonomic classification tables for metagenomic reads were filtered for samples representing an abundance above 1% at the phylum level and genus level. Taxonomic composition was graphed in barplots with ggplot2 [67].

Two non-metric multidimensional scaling (NMDS) were performed to represent taxonomic composition at genus level among 14 metagenomes of microbial mats, one for eukaryotes and one for prokaryotes. Taxonomic counts were pre-normalized using the Trimmed Mean of M-values (TMM) method to correct for possible biases in sequence abundance [68]. Taxonomic assignments were performed at the read level, allowing identification of microbial diversity in both groups. The NMDS analysis was based on the dissimilarity matrix calculated using the Bray-Curtis distance suitable for relative abundance data, with vegdist from vegan package. The final dimensionality of each NMDS was selected considering the stress of the model, ensuring values lower than 0.2 for a good representation of the data in the two-dimensional space. The analyses were carried out in the R statistical environment using the vegan package [66].

The comparative analysis focused on the Fildes Peninsula (MA) and Garwood Valley (DV) samples, considering their sample size and physicochemical characteristics. Using the STAMP platform [69], Welch t-tests were performed to compare the relative abundance of prokaryotes, eukaryotes, and microbial functions at both the reads and gene level. Mantel tests were performed to compare mat composition, environmental variables, and geographic position. Spearman correlations between genus composition and abundance were analyzed in relation

to environmental variables (adjusted p-value FDR < 0.01; rho ≥ |0.75|). The potential metabolic capacities of each metagenome were visualized using a bubble plot created with ggplot2 [67]. Images were subsequently edited with GIMP (https://www.gimp.org/) and Inkscape (https://inkscape.org/) for presentation purposes.

## Results

Microbial mats analyzed in this Antarctic study were sampled between latitude -62.16 S to -78.03 S, encompassing a great extension of West Antarctic regions including Maritime Antarctic (MA), Antarctic Peninsula (AP) and Dry Valleys (DV). The microbial mats sampled presented a multilayer structure, although in some samples it was not easily recognizable. Microbial mats showed a pigmentation of orange and green tones, with an average thickness of 13 ± 7 mm, showing a totally irregular morphology. The analyzed mats develop in the benthos of meltwater streams, forming pigmented biofilms dominated mainly by cyanobacteria and microalgae. At all sites, a discontinuous patchy distribution was observed, where microbial mats were visibly dark green in areas closer to the stream bank, and mats in submerged sections showed orange coloration (Fig 1A-C, S2 Fig1).

### Nutrients in water surrounding

Concentrations of inorganic nutrients (SRP, SRSi, $NO_2^-$, $NO_3^-$, $NH_4^+$, DIN) and the DIN:SRP ratio of surrounding water were analyzed at the 14 sites sampled (Table 1). Concentrations of soluble reactive phosphorus (SRP) ranged from 0.07 μM (Fildes2) to 26.24 μM (Primavera), with a median of 0.365 μM. On the other hand, soluble reactive silicon (SRSi) showed a wide range, from 4.97 μM (Taylor) to 107.9 μM (Garwood10), with a median of 28.005 μM. Dissolved inorganic nitrogen (DIN, which includes $NO_2^-$, $NO_3^-$ and $NH_4^+$) concentrations ranged from 1.18 μM (Taylor) to 413.7 μM (Primavera), with a median of 12.925 μM. Nitrite ($NO_2^-$) and nitrate ($NO_3^-$) concentrations were high in the Esperanza, Primavera and San Martin (AP) samples. The DIN:SRP ratio showed considerable variation, with minimum values of 2.92 (Taylor) and maximum of 140.39 (Potter6), and a median of 51.465. The highest values of this ratio were observed at sites with lower SRP concentrations. In general, the highest DIN concentrations were found at Primavera (AP), which also had the highest ammonium ($NH_4^+$) concentrations (Table 1).

The PCA-biplot including nutrient concentration vectors (SRSi, SRP, $NO_2^-$, $NO_3^-$, $NH_4^+$, DIN and DIN:SRP ratio) revealed a clear separation of the three samples belonging to AP with respect to the rest of the evaluated sites. This separation was particularly remarkable in the Primavera sample towards which the phosphorus (SRP), ammonium ($NH_4^+$), dissolved inorganic nitrogen (DIN) and nitrite ($NO_2^-$) vectors pointed, indicating an association between these nutrients and this specific site. In addition, phosphorus (SRP), ammonium ($NH_4^+$), dissolved inorganic nitrogen (DIN) and nitrite ($NO_2^-$) showed the highest values of quality of representation in the biplot ($cos^2$), suggesting that these nutrients best explain the variation observed in the data. On the other hand, the Fildes2 and Potter6 samples were characterized by the highest values in the DIN:SRP ratio (Fig 2).

### Elemental C, N and P in mats

Carbon (C), nitrogen (N) and phosphorus (P) ratios in the biomass of the microbial mats were analyzed by comparison with Redfield ratios (C:N of 6.625 and N:P of 16). The C:N ratio measured by elemental analysis ranged from 4.75 (Potter6) to 19.97 (Fildes1), with a median of 9.51. Sites Fildes1, Fildes2 and Esperanza had higher C:N ratio values. Only the Potter6 sample, with a C:N value of 4.75, is below the reference value. Regarding the N:P ratio,

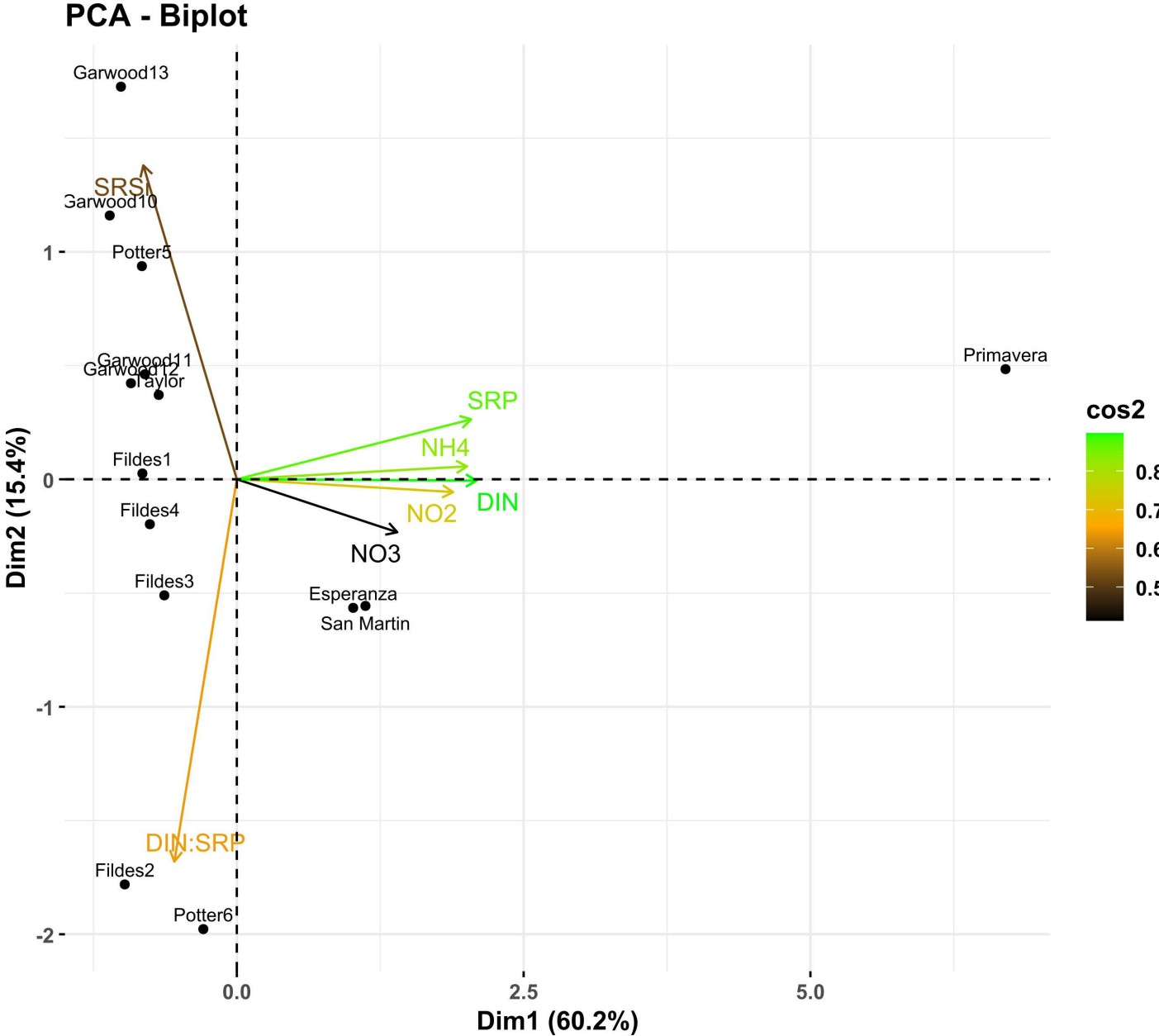

**Fig 2. PCA biplot.** Relationship between sampling sites and nutrient concentration vectors (SRSi, SRP, $NO_2^-$, $NO_3^-$, $NH_4^+$, DIN, and DIN:SRP ratio). The length and direction of the vectors indicate the contribution of each nutrient to the variation in the dataset. The color gradient represents the cos² values, indicating the quality of representation of each nutrient in the ordination.

calculated by Valderrama's method, values ranged from 1.98 (Potter5) to 18.71 (Fildes2), with a median of 6.75. Values below the Redfield N:P ratio of 16 were observed at most sites, for example at Potter5 (1.98), Potter6 (3.59) and Primavera (4.65). Only the sample at Fildes2 (18.71) has a value higher than 16 (Table 1).

## Metagenomic overview

This project generated close to 363.5 million paired end reads, with an average of 24.5 million paired end reads per sample, with a read length of 151 bp. About 80% of the sequenced reads

passed quality control with a Q20 rate above 0.92 and a Q30 rate above 0.83. On average, 8 million paired end reads were assigned per metagenome using Kaiju. A total of 1,118,337 contigs larger than 1000 bp were assembled, of which 659,638 were assigned to prokaryotes and 30,256 to eukaryotes with the remaining 427,609 as unclassified. A total of 861,591 prokaryotic coding sequences (CDS) and 148,560 eukaryotic CDS were predicted. In addition, 6,055,548 open reading frames (ORFs) of unclassified contigs were predicted (S1 Summary statistics).

## Community composition

Based on the Kaiju classification, Antarctic microbial mat metagenomic reads were assigned to Bacteria (93.45%), Eukarya (3.32%), Archaea (0.09%), and Viruses (0.09%). The average abundance of the main phyla composing Antarctic microbial mats included Bacteroidota (34.61%), Pseudomonadota (28.22%), Cyanobacteriota (18.60%), Verrucomicrobiota (3.02%), Bacillariophyta (2.14%), Planctomycetota (1.81%), Acidobacteriota (1.60%), Actinomycetota (1.55%), Bacillota (0.79%) and Chloroflexota (0.93%) (Fig 3A). At the genus level, taxa depicting an overall abundance higher than 1% are mainly related to phylum Bacteroidota: *Flavobacterium, Pedobacter, Spirosoma, Hymenobacter, Arcicella, Flexibacter*; Pseudomonadota: *Polaromonas, Sphingomonas and Sandarakinorhabdus*. Cyanobacteriota genera include *Pseudanabaena, Alkalinema, Leptolyngbya, Nostoc, Chamaesiphon and Microcoleus* (Fig 3B). There is a large number of reads that cannot be assigned taxonomically (62.3%) suggesting need further exploration.

Mats from Fildes1, Fildes2, Fildes3, Fildes4, Potter5, Potter6 and Esperanza have approximately 4% of sequences associated with the phylum Bacillariophyta. In contrast, the Garwood12 sample presents 6.36% of sequences associated with the phylum Chloroflexota, while in the rest of the samples, Chloroflexota is represented in less than 1%. The composition of the San Martin sample has a distinct composition in relation to the most abundant phyla. The San Martin metagenome shows a high proportion of sequences associated with the phylum Bacteroidota (69.88%), less than 1% of sequences associated with the phylum Cyanobacteriota, 4.75% of sequences associated with Actinomycetota and 2.81% of sequences associated with the phylum Chlorophyta. In addition, the proportion of sequences associated with the genera Flavobacterium (25.19%), Spirosoma (6.91%), Hymenobacter (6.70%) and Pedobacter (6.24%), all belonging to the phylum Bacteroidota, are outstanding in the San Martin microbial mat compared to the rest of the samples (Fig 3).

The eukaryotic phyla present in Antarctic microbial mats with a relative abundance greater than 1% included Bacillariophyta and Chlorophyta, representing 1.72% and 1.24% of the total number of reads. The taxonomic assignment with EukDetect identified *Plectus murrayi*, a nematode in mats from MA (Fildes1, Fildes2) and DV (Garwood12, Garwood13). The tardigrade *Hypsibius dujardini* and fungi *Protomyces sp*. C29, *Glaciozyma antarctica* were also identified, the latter in Esperanza and Primavera (AP). Interestingly, the flightless fly *Belgica antarctica* was identified with several genomic markers (S1 Summary statistics). In addition, we found several microeukaryotes, such as ciliates, rotifers, nematodes, fungi, microalgae from the partial annotation of the 18S rRNA gene. Based on the annotation of partial 18S rRNA gene sequences, we performed the phylogenetic reconstruction of *Adineta vaga* (S2 Fig 5).

OTU assignments were based on 14 ribosomal proteins as unique markers. A total of 9,414 OTUs, taxonomically assigned to species level, were detected in the metagenomes of microbial mats. Diversity indices averaged 4.98 +/- 0.83 for Shannon index and 0.94 +/- 0.05 for Simpson index (S1 Summary statistics). A comparison of NMDS ordination analyses based on Bray-Curtis distance was performed at the genus level for the 14 metagenomes, for both prokaryotes and eukaryotes. The NMDS analysis of prokaryotes presented a stress value of 0.079,

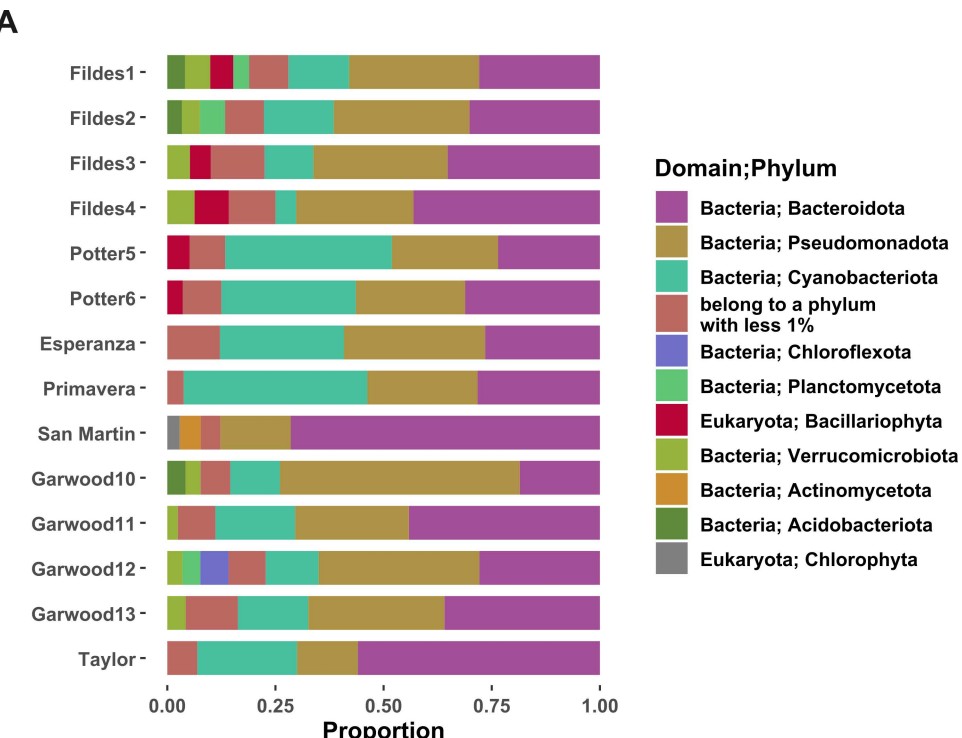

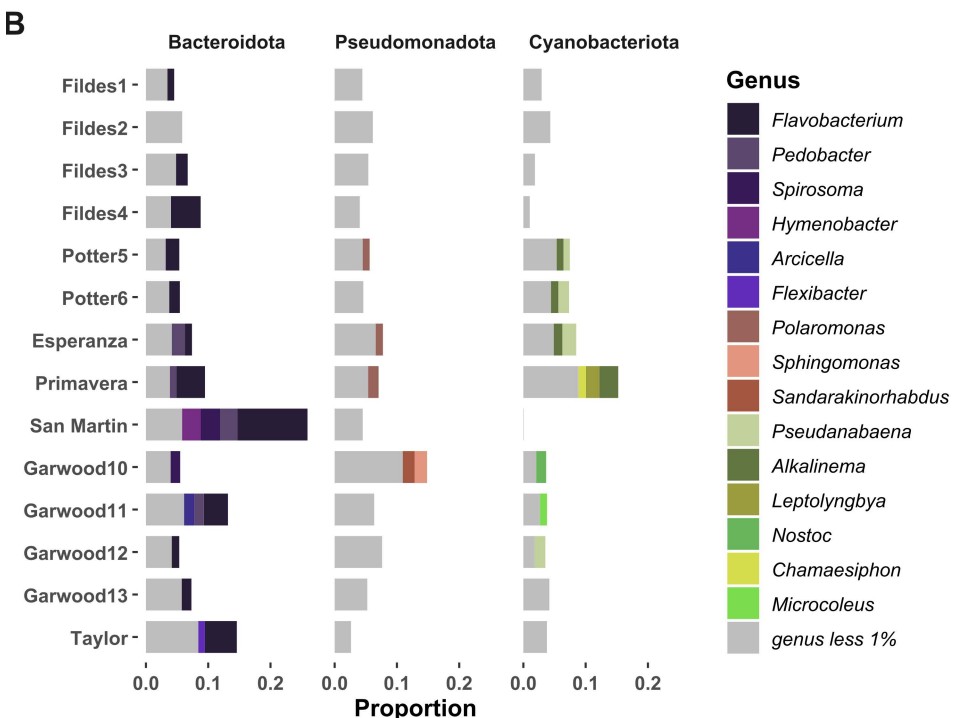

**Fig 3. Taxonomic composition of Antarctic microbial mats.** Abundances at read level greater than 1%. **A)** The phyla with the highest abundance in the microbial mats are Bacteroidota, Pseudomonadota, and Cyanobacteriota, with 80% +/- 10%, and Bacillariophyta in the mats of Maritime Antarctica. **B)** Most abundant genera in microbial mats.

while that of eukaryotes showed a lower stress value of 0.053. In the case of prokaryotes, no clear separation between samples was observed, evidencing a dispersed distribution. On the other hand, the NMDS ordination of eukaryotes revealed a separation between the MA and DV samples, although the samples corresponding to AP showed an overdispersion in both ordinations (S2 Fig2).

## Comparison of geographically distant mat samples

To gain insight into the differential composition between mats among samples from Fildes (MA) and Garwood (DV), Welch's t-test were performed and STAMP analyses showed differential abundance (at read level) in the mats for phylum Bacillariophyta, genus *Phaeodactylum, Fistulifera, Fragilariopsis, Pseudo-nitzschia, Thalassiosira, Halamphora* (Fig 4A, S2 Fig 3), which are significantly more abundant in the marine-influenced sites of Fildes (MA) when compared to Garwood (DV) mats. In addition, at the gene level we found that there is a difference (Welch's t-test p < 0.01) in eukaryotic gene abundance between samples from Fildes (MA) and Garwood (DV). Gene prediction of eukaryotes revealed a great diversity, with significant variations between samples. This genomic information contributes to current knowledge of the role of eukaryotes in microbial mats of Antarctica. The functional classification suggests the importance of Bacillariophyta and unclassified Eukaryota, followed by Ciliophora, Chlorophyta, unclassified Fungi, unclassified Metazoa, Streptophyta and Oomycota. At a smaller scale, Apicomplexa, unclassified Viridiplantae, Basidiomycota, Euglenozoa, Haptophyta, Chytridiomycota and Cercozoa are also present in Antarctic mats. Further, the COG-based classification of annotated genes (Fig.4B) showed the following categories as relevant in mat composition: S (function unknown), O (post-translational modification, protein turnover, chaperones functions), T (signal transduction), L (replication and repair) and J (Translation).

We found no differences in prokaryote composition (read level) in metagenomes, nor in GPM for COGs (genes per million) at the functional level for the most abundant phyla (Bacteroidetes, Proteobacteria, Cyanobacteria, Verrucomicrobia, Firmicutes, Actinobacteria, Planctomycetes, Gemmatimonadetes, Acidobacteria, and Chloroflexi) between the Fildes and Garwood samples, according to Welch's t-test (p > 0.05) (S2 Fig4).

## Discussion

In this study, we present data for dissolved nutrients, microbial biomass (C, N, and P), taxonomic and functional profiles of microbial mats in West Antarctica sites (Fildes Peninsula and Potter Cove in King George Island (South Shetland Islands); Argentinian scientific bases along the Antarctic Peninsula, and Garwood and Taylor Valley in McMurdo Dry Valleys). Although cyanobacterial microbial mats are a ubiquitous component of benthic meltwater Antarctic systems, this study analyzes microbial mats in different environments in a large geographical extension. The results indicate nutrient variability in the sampled sites, nitrogen limitation in most microbial mats, and high concentrations of DIN in the AP samples. In addition, results indicate that the mats from MA islands, rocky shores of the AP, and valleys of DV regions are diverse and dominated by Bacteria. However, eukaryotes introduce significant variation in the composition of microbial mats. These results agree with previous studies regarding bacterial diversity and dominant bacterial phyla [7,14,70,71]. Microeukaryotes that had not been described in metagenomic analysis are reported and their essential role in the functioning of microbial mats, such as consumers in food webs [23,24,72] and productivity [73, 74] is suggested. These results are significant because they contribute additional knowledge about the benthic microbial mats that develop in West Antarctica, particularly in MA

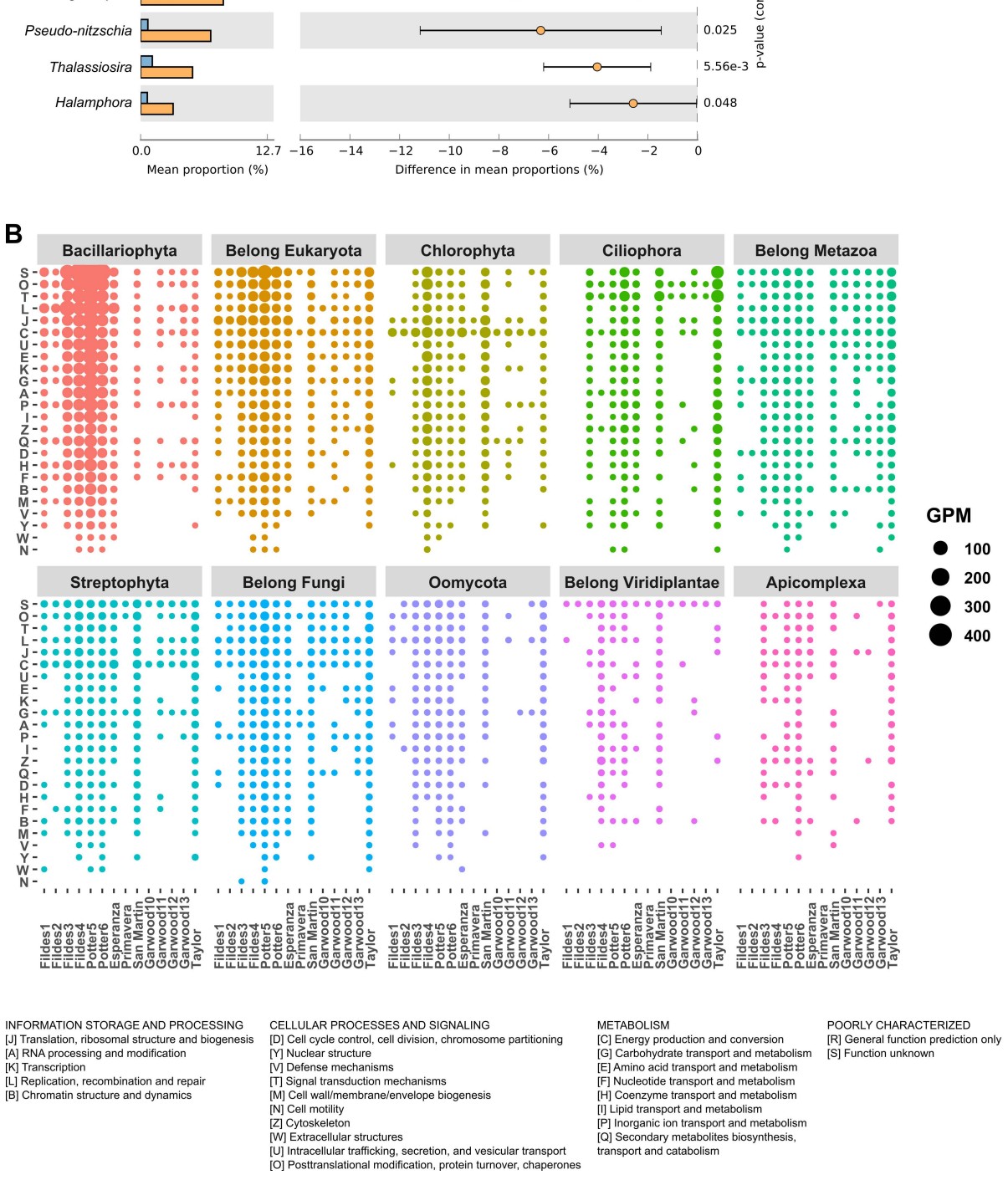

**Fig 4. Eukaryotic proportion comparison and gene distribution between metagenomes.** Eukaryotic gene counts are shown for the 14 microbial mats, organized by the top ten eukaryotic phyla with the highest number of genes. The figure reveals the specific distribution of eukaryotic genes in the communities, highlighting the variation in gene abundance for the differentially abundance in phylum

Bacillariophyta. (A) the difference between mean proportions (read level) for six genera of the phylum Bacillariophyta. The figure highlights that Bacillariophyta are differentially abundant in samples from Fildes (MA) compared to Garwood (DV). (B) Distribution of eukaryotic genes as a function of gene counts per million reads (GPM) and Clusters of Orthologous Genes (COGs) categories.

and AP, regions with an accelerated increase of moss dominated vegetation cover (0.424 Km2 yr-1) in response to raising temperatures [75] and ecosystem changes due to warmer, wetter conditions and higher moisture availability [76].

This study, performed in environments with diverse climatic and environmental conditions, has revealed differences in nutrient availability among the studied sites in West Antarctica. The highest concentrations of phosphorus (SRP), ammonium ($NH_4^+$), and dissolved inorganic nitrogen (DIN) were recorded in Primavera (AP), notably higher than the rest of the sampled sites. This high nutrient concentration is likely related to the input of organic matter from local fauna. Almela et al. [77] have shown how bird and mammal activity on the Antarctic Peninsula can significantly alter the composition and structure of microbial mats. By contributing organic matter, these animals can modify the C/N and N/P ratios of the ecosystem, potentially favoring the accumulation of ammonium ($NH_4^+$) and thereby influencing nutrient dynamics. Similarly, a study by Wang et al. [78] in Victoria Land evaluated the impact of nitrogen ($^{\delta 15}N$) through penguin guano. Penguin activities introduce nutrients such as nitrogen and phosphorus into the ecosystem. Microbial mats near penguin colonies show higher nitrogen enrichment, as indicated by $^{\delta 15}N$ values. Nitrate ($NO_3^-$) concentration is high in water samples from Esperanza, Primavera, and San Martín (AP) sites. The results of Camacho et al. [79] at Byers Peninsula (MA), where ammonium ($NH_4^+$) was the preferred nitrogen source in microbial mats, suggest that in San Martin processes could be occurring that limit nitrate utilization by microorganisms, which would explain its accumulation. In addition, the composition and abundance of the San Martin microbial mat are likely affected by toxicity and osmotic potential at high nitrate ($NO_3^-$) concentrations [80]. However, our study did not find a correlation between nutrients and community abundance, nor an association with diversity. Samples from the Fildes2 and Potter6 sites in MA were characterized by the highest DIN:SRP ratio values, which could be related to phosphate limitation (SRP) at these sites. Interestingly, gene annotation indicates that both metagenome samples possess *pho* and *pst* genes as a strategy used in bacteria to scavenge phosphate in a P-limited environment [81–83].

The C/N and N/P ratios in the microbial mats suggest possible nitrogen limitation when compared to Redfield ratio reference values (C/N = 6.625 and N/P = 16). However, it is important to note that acidification was not performed to the samples to remove potentially present carbonates, which could have overestimated the carbon content. Despite this, the percentages of nitrogen and phosphorus in biomass obtained by Valderrama's method reinforce the nitrogen limitation hypothesis. Antarctic freshwater environments are considered oligotrophic; microbial mats develop adaptive mechanisms to cope with nitrogen limitation, including nitrogen immobilization, and nitrogen fixation [84, 85]. No correlations were observed between C/N—N/P ratios and microbial composition, abundance, or diversity, which is consistent with previous studies conducted in the McMurdo Dry Valleys [86]. Although nitrogen limitation may influence mat biomass, other factors, such as environmental conditions or interspecies competition, may be more relevant in structuring the microbial community. The lack of correlation also suggests that microbial diversity and composition could be controlled by variables independent of biomass, such as temperature, light availability, or other energy sources [7,87]. Future research must include a broad view encompassing both oligotrophic and those with varying degrees of nutrient accumulation. This approach will provide

a comprehensive understanding of how the cumulative effects of nutrient cycling processes influence the structure and functioning of microbial mat communities.

Metagenomics has proven essential to unraveling the composition and biological functions of microeukaryotes, a group with low abundance in Antarctic mats. Microbial mats in Western Antarctica exhibit a taxonomically consistent composition, with bacterial dominance and notable common phylum Bacteroidota, Pseudomonadota, and Cyanobacteriota [7,14,15,86]. However, there is also a diversity of microeukaryotes [88–90], archaea and viruses to a lesser extent [91, 92]. Interestingly, we observed an association between sites from eukaryote-related ordination and found a difference in microbial mat composition in genera of the phylum Bacillariophyta at Fildes Peninsula sites in MA compared to Garwood Valley sites in DV. Diatoms are among the most abundant microalgae in terrestrial and freshwater ecosystems in the Antarctic and sub-Antarctic regions [32]. Diatom communities are sensitive to salinity, pH, temperature, and other physical variables [93] with conductivity being among the physical variables that best explain diatom community structure [94]. A high affinity in diatom composition explained by the electrical conductivity of water in lakes in the northern Antarctic Peninsula region has been observed [32]. The difference in diatom abundance in the Fildes Peninsula site mats may be due to conductivity.

Our study represents a pioneering advance by employing a shotgun sequencing metagenomics approach to characterize and compare West Antarctica microbial mats, providing detailed insight into the biological diversity of this unique region. Although metagenomics has been widely used in microbial communities, there has been a predominant trend towards analyzing prokaryotes. This is reflected in databases such as GTDB, which have principally bacterial and archaeal genomes generated from metagenomic samples [95]. Our work contributes to closing this gap by highlighting the presence and diversity of microeukaryotes in an Antarctic environment. Identifying extremely stress-tolerant organisms in Antarctic mats underscores the capacity of these ecosystems to support life forms adapted to extreme conditions. The presence of *Glazyozyma antarctica*, a fungus with antifreeze genes [96], *Plectus murrayi*, a nematode adapted to harsh environments [97], *Belgica antarctica*, an endemic flightless fly [98], and *Hypsibius dujardini*, a tardigrade resistant to extreme conditions in Antarctic soils [99,100], highlights the functional and adaptive diversity of microeukaryotes to this unique environment.

Results show low molecular variation in the V4 region of the 18S rRNA gene of *Adineta vaga* (Bdelloidea, Rotifera) at different localities in West Antarctica. The consistent presence of *A. vaga* in Antarctic mats suggests a unique adaptability to this extreme environment [101]. However, although the V4 region of the 18S rRNA shows low variability, it is possible that other molecular markers, such as mitochondrial genes or whole genome analyses, may reveal more complex patterns of genetic diversity. This type of analysis would expand our understanding of the biogeography and evolution of *A. vaga* on the Antarctic continent, providing essential information on the dynamics of isolation and connectivity in different Antarctic regions [102]. These findings highlight the need for more detailed investigations into the ecology and physiology of microeukaryotes in Antarctic environments. Our results support the idea that Antarctic microbial mats are not only hotspots of diverse microorganisms but could also be considered refugia of Antarctic diversity.

Here we contribute to the understanding of benthic microbial mats in West Antarctica on a considerable geographic scale. We hope our results will provide a foundation for future studies, particularly considering the variability and uniqueness of environmental conditions at each sampling site. Culture-independent approaches merit further applications to study the diversity of microeukaryotes, archaea, and viruses in microbial mats, as these groups remain understudied. Finally, documenting diversity in an ecosystem as vulnerable to global

warming as Western Antarctica holds critical value for assessing and preserving its ecological resilience.

## Supporting information

**S1 Summary statistics. Metagenomic summary processed.**
(XLSX)

**S2 Supplementary figures.**
(PDF)

## Acknowledgments

Research presented here is dedicated to the memory of S. Craig Cary, an intrepid microbial explorer of Antarctica and its Dry Valleys who was an inspiration for many. This paper is part of the requirements for obtaining a Doctoral degree at the Posgrado en Ciencias Biológicas, UNAM (RAMJ). RAMJ thanks Consejo Nacional de Humanidades, Ciencias y Tecnología (CONAHCyT) for a graduate studies scholarship. PMV-C acknowledges an Antarctic Science International Bursary Early Career award.

## Author contributions

**Conceptualization:** Ricardo A. Mercado-Juárez, Patricia M. Valdespino-Castillo, Martín Merino Ibarra, Silvia Batista, Walter Mac Cormack, Lucas Ruberto, Edward J Carpenter, Douglas G Capone, Luisa I. Falcón.

**Data curation:** Ricardo A. Mercado-Juárez, Patricia M. Valdespino-Castillo, Luisa I. Falcón.

**Formal analysis:** Ricardo A. Mercado-Juárez, Patricia M. Valdespino-Castillo, Luisa I. Falcón.

**Funding acquisition:** Patricia M. Valdespino-Castillo, Martín Merino Ibarra, Silvia Batista, Walter Mac Cormack, Edward J Carpenter, Douglas G Capone, Luisa I. Falcón.

**Investigation:** Ricardo A. Mercado-Juárez, Patricia M. Valdespino-Castillo.

**Methodology:** Ricardo A. Mercado-Juárez, Patricia M. Valdespino-Castillo, Lucas Ruberto, Luisa I. Falcón.

**Project administration:** Luisa I. Falcón.

**Resources:** Luisa I. Falcón.

**Software:** Ricardo A. Mercado-Juárez.

**Supervision:** Patricia M. Valdespino-Castillo, Luisa I. Falcón.

**Validation:** Ricardo A. Mercado-Juárez, Patricia M. Valdespino-Castillo, Luisa I. Falcón.

**Visualization:** Ricardo A. Mercado-Juárez, Luisa I. Falcón.

**Writing – original draft:** Ricardo A. Mercado-Juárez, Patricia M. Valdespino-Castillo, Luisa I. Falcón.

**Writing – review & editing:** Ricardo A. Mercado-Juárez, Patricia M. Valdespino-Castillo, Silvia Batista, Lucas Ruberto, Luisa I. Falcón.

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
