## [Decision Letter · Decision Letter 0]

28 May 2024

PONE-D-24-02443
What defines a photosynthetic microbial mat in western Antarctica?
PLOS ONE

Dear Dr. Falcón,

Thank you for submitting your manuscript to PLOS ONE. After careful consideration, we feel that it has merit but does not fully meet PLOS ONE’s publication criteria as it currently stands. Therefore, we invite you to submit a revised version of the manuscript that addresses the points raised during the review process.
I dis also my personal review of the manuiscript and I agree with the reviewer. Particularly, please give more details about the mat features and consider the methodological issues raised by the reviewer. 

We look forward to receiving your revised manuscript.

Kind regards,

Andrea Franzetti

Academic Editor

PLOS ONE

Journal Requirements:

[This paper is part of the requirements for obtaining a Doctoral degree at the Posgrado en Ciencias Biológicas, UNAM of RAMJ. Financing was granted by AMEXCID, AUCI and DNA to WM, SB, LIF, PMV-C. DGC thanks NSF for sustained support. We thank the Consejo Nacional de Humanidades, Ciencias y Tecnologías (Conahcyt) for the support through a graduate scholarship to RAMJ.]

 [Financing was granted by AMEXCID, AUCI and DNA to WM, SB, LIF, PMV-C. DGC thanks NSF for sustained support.]

 [Financing was granted by AMEXCID, AUCI and DNA to WM, SB, LIF, PMV-C. DGC thanks NSF for sustained support.].  

6. Please provide a complete Data Availability Statement in the submission form, ensuring you include all necessary access information or a reason for why you are unable to make your data freely accessible. If your research concerns only data provided within your submission, please write "All data are in the manuscript and/or supporting information files" as your Data Availability Statement.

Reviewers' comments:

Reviewer's Responses to Questions

**Comments to the Author**

1. Is the manuscript technically sound, and do the data support the conclusions?

Reviewer #1: Yes

2. Has the statistical analysis been performed appropriately and rigorously? 

Reviewer #1: Yes

3. Have the authors made all data underlying the findings in their manuscript fully available?

Reviewer #1: Yes

4. Is the manuscript presented in an intelligible fashion and written in standard English?

Reviewer #1: No

5. Review Comments to the Author

Reviewer #1: This is interesting contribution shows the diversity of microorganisms both euks and proks in microbial mats from a geographical gradient of freshwater ecosystems. The authors use metagenomic approach to obtain the diversity of the mats as well as the functionality of each one. The main result is that while the prokaryotic community is very similar in all mats, the eukaryotic community provides clear differential characteristics at different latitudes

The authors cluster a number of microbial mats collected in geographical locations and cluster them as representative of the same region. They do not consider that chemically and biologically they are not similar and thus including those ‘outliers’ in the study can bias the results. A clear example is SM mat which is clearly different from the rest and most likely not associated to the latitude, but to the chosen one. At that latitude ‘normal’ mats have been described and published elsewhere. This is an example, but other cases are obvious as well. Looking at the biological composition of the microbial mats investigated it is quite obvious that are not comparable among them. For instance, SM has only 0.19% of cyanobacterial reads, while other in the same clustering region (Peninsula) had 2 orders of magnitude more cyanobacteria, for example.

Obviously the mats are different in the closeby locations (Table 2) and cluster them by geographical region may not be appropriate. Looks like Primavera mats are influenced very much by animals, with high P and ammonium concentrations, and similar situation is observed in other sites. It is difficulty to cluster things that are not similar., although located close by

Having conductivity and pH values of the surrounding waters cold have helped choosing comparable ecosystems. I would recommend also to use ammonium to discard those mats subjected to animal colonies influences (please see Almela et al 2022). The proximity to animal colonies produces communities quite different to others at the same latitude without animal influence.

I miss a more precise description of the mats: thickness, color, aspect and so on And perhaps a plate in suplementary material with pictures of the microbial mats collected would be useful for the readers. More precise description of the sites is needed, not so much the geographical region but the precise sites.

Some precise comments:

Legends for supplementary material are required, some are difficult to understand

In the elemental analyses there was a preacidification to eliminate the calcium carbonate from the samples?. The Garwood samples show a peculiar C/N an C/P behaviour, that could be derived from a high calcium carbonate content (as evidenced by the high Ca content in those samples). Without preacidification is very hard to do an interpretation of the results of elemental composition because minerals and biominerals accumulated in the microbial mat might not be considered part of the biomass.

Are the 14 first lines in the intro relevant for the paper?

Lines 70-73 in introduction are redundant just say are all over

Lines 74-75 also in permanently ice covered lakes

A bit poor the intro, there is much more literature available covering euks

The format of the tables is quite uneasy.

What is the asterisks in table 2?

I think figure 3 and 4 are changed. In figure of the taxonomic composition it is very clear that SM mat is completely different, how this difference influence the results?

Lines 168-170, this is well known. Microbial mats in Antarctica include a quite diverse consortium of eukaryotes

line 189, I guess some words are missing

Figure 6 is not clear to me and it is not properly explained in the text.

What is the contribution of Figure 7 to the leading line of the paper?, most of the metabolic pathways searched are present in most microbial mats, as expected.

Lines 260-264, I do not agree. The composition of euk in microbial mats has been discussed since 1910 in the Murray papers and since then in last 10-15 years a number of papers (some not referenced) are available.

Lines 268-272, I do not understand the discussion about humidity, I guess this could be deferred towards liquid water availability, since humidity is not very relevant for aquatic ecosystems. In fact, I think the LIQUID water availability could be relevant for the discussion since in the peninsula area with much higher precipitation is also warmer and very frequently microbial mats might be dry (at the end of summer), in the MCM region it is much less precipitation but ice is more permanent and dryness can also be experimented by microbial mats because of lack of liquid water due to ice formation. The authors may want to elaborate on that

Lines 299, why are they psicrophilic extremotolerant? Who said that? Any reference?

Sentence in Line 346 347 is very similar to 318-321

Line 359 about filtering is not clear to me

lines 375 and 376 nitrite and nitrate are wrong

Line 384 I do not understand that sentence

S1B where is ammonium?

Is last paragraph in the discussion new?, necessary?, relevant?

6. PLOS authors have the option to publish the peer review history of their article (what does this mean?). If published, this will include your full peer review and any attached files.

Reviewer #1: No

---

## [Author Response · Author response to Decision Letter 1]

19 Nov 2024

Dear Reviewer:

We sincerely appreciate your feedback and suggestions for enhancing our manuscript. Below is a summary of our key changes to the paper.

 1. Grouping of microbial mats by geographic region: According to your suggestion, we reconsidered grouping the samples by geographic regions due to the chemical and biological variations among the mats. We note that specific samples, such as the one from San Martin (SM), exhibit unique characteristics, such as the low proportion of cyanobacteria (0.19%) compared to other samples from the same region (two orders of magnitude more in some). These differences may be due to specific local factors rather than being associated with latitude. Therefore, we discarded the grouped analysis and focused on studying the mats individually, which allowed a more precise interpretation of each site.

 2. Influence of animal colonies and elevated nutrients (NH₄⁺ and SRP): We appreciate the suggestion to use ammonium as a criterion to identify the influence of animal colonies on microbial mats. We incorporate a detailed analysis of ammonium and soluble reactive phosphorus levels, considering that these elevated concentrations at sites such as the Antarctic Peninsula could bias the results. In addition, a direct comparison between the Fildes Peninsula and Garwood Valley mats allowed for more robust and representative data.

 3. Accurate description of the microbial mats: As you recommended, we added a detailed description of the microbial mats, including aspects such as thickness, color, and appearance. We also included photographs of the mats in the supplementary material, which we hope will provide a better visual understanding of the characteristics of each sample and facilitate the interpretation of the data.

 4. Incorporation of PCA-biplot for nutrient characterization: To reinforce the analysis of the chemical characteristics of water that inhabits the mats, we included a PCA-biplot graph that allows a clear visualization of the nutrient characterization at each site. This analysis helps to illustrate the variations between samples in more detail and facilitates the comparison of ecosystems based on their chemical and biological properties.

 5. Organization of the manuscript structure: Following the recommendations, we adopted a more traditional organization, with a clear section for the introduction, materials and methods, results, and discussion. We believe this scheme will improve the document's clarity and allow readers to find the information more structured.

 6. Discard the metabolic pathway gene analysis: We opted to remove this section due to the high levels of false positives detected in the metabolic pathway gene analysis. Eliminating the analysis reduces the risk of confounding interpretations and ensures greater accuracy in the results. Curating these data would have required a laborious and extensive process, which could have been more practical in the setting of this study.

 7. Comparison between Fildes Peninsula and Garwood Valley samples: Based on your comments, we decided to focus the comparative analysis on the Fildes Peninsula (MA) and Garwood Valley (DV) samples, as these sites provided a more balanced data set suitable for statistical analysis.

 8. Reorganization of figures: We reduced the number of principal figures in the manuscript and organized them in the supplementary file, simplifying the manuscript's structure and allowing for smoother reading.

Point-by-point comments and responses to Reviewer I:

Comment #1: Legends are needed for the supplementary material, some are difficult to understand.

Response: We appreciate the suggestion and have made the requested adjustments. We have labeled the supplemental file to make it readable and understandable. It can be found in the supplementary file S1.

Comment #2: Pre-acidification was performed in the elemental analyses to remove calcium carbonate from the samples..... The Garwood samples show a peculiar C/N and C/P behavior, which could derive from a high calcium carbonate content (as evidenced by the high Ca content of those samples). Without pre-acidification it is very difficult to make an interpretation of the elemental composition results because the minerals and biominerals accumulated in the microbial mat might not be considered part of the biomass.

Response: We appreciate your noticing the potential overestimation in the C/N and C/P ratios. In the discussion (lines 454-456), we have clarified that pre-acidification was not conducted, and that C overestimation is suspected due to the presence of carbonates.

Comment #3: Are the 14 first lines in the intro relevant for the paper?

Lines 70-73 in introduction are redundant just say are all over

Lines 74-75 also in permanently ice covered lakes

A bit poor the intro, there is much more literature available covering euks

Response: We acknowledge that the introduction required more depth. To enhance this section, we have included information on how climate change impacts polar regions, focusing mainly on its effects on microbial mats and the role of eukaryotes in these ecosystems.

Comment #4: The format of the tables is quite uneasy. What is the asterisk in Table 2?

Response: We agreed that grouping the data by zones was not appropriate due to each site's environmental variability. We now present in Table 1 the geographic, nutrient, and biomass data for each mat sampled (lines 274-280). Asterisks represent significant differences in the Kruskal-Wallis test.

Comment #5: I think figure 3 and 4 are changed. In figure of the taxonomic composition it is very clear that SM mat is completely different, how this difference influence the results?

Response: Thank you for your feedback. We have corrected the order of Figures 3 and 4 and are now presenting the taxonomic composition first (Figure 3). To enhance readability, the bars in Figure 3 are now displayed horizontally.

The NMDS (Non-metric Multidimensional Scaling) ordering has been moved to Supplementary Figure S2. However, organizing the data by site while separating prokaryotes and eukaryotes remains essential for understanding the differences in microbial mat composition, particularly in the case of the San Martin mat.

As you noted, the composition of the San Martin mat is significantly different from that of the other samples. Therefore, conducting a pooled analysis by zone could potentially obscure these differences.

Comment #6: Lines 168-170, this is well known. Microbial mats in Antarctica include a quite diverse consortium of eukaryotes.

Response: We appreciate the reviewer's comment. We have modified the text in community composition (318-330) to describe the results only, eliminating any interpretative or redundant elements by the style expected in the results section.

Comment #7: line 189, I guess some words are missing

Response: The reviewer has correctly pointed out that line 189, "The results indicate that while (Fig. 4A) shows all microbial communities, section (Fig. 4B) reveals a more pronounced separation between the regions, highlighting the significant influence of eukaryotic microorganisms on the community structure of microbial mats in western Antarctica."', needs more clarity. We wanted to express that eukaryotic microorganisms play a fundamental role in the structure of microbial mats, contributing significantly to their diversity and organization.

Comment #8: Figure 6 is not clear to me and it is not properly explained in the text.

Response: We appreciate the reviewer's comment. Figure 6 presents a phylogenetic tree constructed from partial 18S rRNA gene sequences of Adineta vaga (Rotifera) obtained from 10 metagenomes of Antarctic microbial mats. This figure shows the low genetic variability observed in the V4 region of this rotifer in the different samples analyzed.

We understand that the figure may require further explanation. The Materials and Methods (lines 207-210), Results (lines 359-362), and Discussion (lines 507-518) sections provide a more complete description of the methods used and results obtained. Supplementary Figure 5 (S2) shows the phylogeny of the sequences related to A. vaga.

Comment #9: What is the contribution of Figure 7 to the leading line of the paper?, most of the metabolic pathways searched are present in most microbial mats, as expected.

Response: We appreciate the reviewer's comment. Figure 7 was intended to illustrate the presence of key metabolic pathways in Antarctic microbial mats, such as nitrogen fixation coupled to oxygenic photosynthesis and sulfur, carbon, and methane cycling. These processes confirm the fundamental role of these ecosystems in global biogeochemical cycles.

However, we decided to eliminate the analysis of metabolic pathway genes due to the high number of false positives obtained. This problem and the time required for data curation could compromise the reliability of our results and hinder their interpretation.

Despite this limitation, our results suggest that Antarctic microbial mats exhibit remarkable metabolic capacity, making them model systems to study their role in biogeochemical cycling in extreme environments. Future research could further explore the functional diversity of these ecosystems using more robust metagenomic approaches.

Comment #10: Lines 260-264, I do not agree. The composition of euk in microbial mats has been discussed since 1910 in the Murray papers and since then in the last 10-15 years a number of papers (some not referenced) are available.

Response: We appreciate the reviewer's comment. It is true that the composition of eukaryotes in microbial mats has been the subject of numerous studies since the early 20th century, such as Murray's pioneering work. However, our study brings a new perspective by addressing this question from a metagenomic (shotgun) approach.

By analyzing metagenomes of Antarctic microbial mats, we have identified a diversity of microeukaryotes that, to our knowledge, had yet to be described from this approach.

A more detailed discussion of the novelty of our findings and how they compare with previous studies can be found on lines 486-499.

Comment #11: Lines 268-272, I do not understand the discussion about humidity, I guess this could be deferred towards liquid water availability, since humidity is not very relevant for aquatic ecosystems. In fact, I think the LIQUID water availability could be relevant for the discussion since in the peninsula area with much higher precipitation is also warmer and very frequently microbial mats might be dry (at the end of summer), in the MCM region it is much less precipitation but ice is more permanent and dryness can also be experienced by microbial mats because of lack of liquid water due to ice formation. The authors may want to elaborate on that

Response: We thank the reviewer for his valuable observation on the importance of liquid water availability in Antarctic microbial mats. We agree that this factor, along with others such as temperature, salinity, and pH, can significantly influence the structure and function of these communities.

While our discussion focused on the role of conductivity as an explanatory factor for diatom distribution, we recognize that liquid water availability is a complementary factor that could help explain the observed variations in community composition. Previous studies in freshwater aquatic ecosystems in Antarctica have shown that conductivity is closely related to diatom diversity and abundance.

However, it is important to note that our study needs complete data on environmental variables such as pH, temperature, and conductivity at all sampling sites. This limitation prevents us from performing a more detailed analysis of the relationship between these variables and the distribution of diatoms. Despite this restriction, the results suggest that conductivity could be an essential factor in future studies.

Future studies could address this issue by collecting more complete data on environmental conditions at the sampling sites. In addition, the use of more sophisticated statistical models would allow for a more accurate assessment of the integral effect of multiple environmental variables on the structure of microbial mat communities, especially diatom communities.

Comment #12: Lines 299, why are they psicrophilic extremotolerant? Who said that? Any reference?

Response: We thank the reviewer for his thoughtful observation. He is right to point out that "psychrophilic extremotolerant" may be ambiguous and lack a solid basis in the scientific literature.

Initially, we considered this term to emphasize the ability of microorganisms in Antarctic microbial mats to survive and thrive in extreme cold conditions. However, we recognize that this classification is inaccurate and can lead to confusion.

Therefore, we have decided to modify this statement. In lines 516 to 518 of the discussion, we now state, "Our results support the idea that Antarctic microbial mats are not only hotspots of diverse microorganisms but could also be considered refugia of Antarctic diversity." This reformulation more accurately captures these ecosystems' importance as biodiversity reservoirs in an extreme environment.

Comment #13: Sentence in Line 346 347 is very similar to 318-321

Response: We thank the reviewer for his detailed comment. We have taken your comment into account and have made the following modifications to improve the clarity of the manuscript:

We rephrase the sentences in lines 100-105. Instead of repeating the information presented in lines 318-321, we restructured the sentences to avoid redundancy and improve the flow of the text.

We have included additional information on the sampling sites: We have included a more detailed description of the sampling sites (113-134). This information better contextualizes our results and facilitates understanding of the differences observed between sites.

These changes effectively address the reviewer's observation and contribute to a more precise and concise presentation of the sampled sites.

Comment #14: Line 359 about filtering is not clear to me

Response: We thank the reviewer for his thoughtful comment. We have modified the wording of line 359 to provide a more precise description of the filtering process.

The sentence now reads, "Water samples for nutrient analysis were collected in polypropylene acid-washed bottles, in triplicates, and filtered through nitrocellulose membranes 0.45 and 0.22 µm (HA Millipore™)."

We used sequential filtration, first through a 0.45 μm membrane and then through a 0.22 μm membrane. This double filtration aimed to retain particles and microorganisms of different sizes, thus ensuring that only the nutrients dissolved in the water were analyzed. Nitrocellulose is a commonly used material in the filtration of water samples for nutrient analysis, as it effectively retains particles without adsorbing the nutrients of interest.

By providing this additional information, we hope the reviewer will better understand the purpose and methodology of our filtration.

Comment #15: lines 375 and 376 nitrite and nitrate are wrong

Response: We thank the reviewer for identifying the error on lines 375 and 376. Indeed, the chemical formula for nitrite and nitrate had been reversed.

We have corrected this error on lines 154 to 160, where the analytical procedure is detailed. It is now clearly stated that nitrate (NO₃-) and nitrite (NO₂-) concentrations were determined using a Skalar San Plus continuous-flow autoanalyzer following the standard methods.

In addition, we have verified all data and tables to ensure that the chemical formulas and corresponding values are correct.

We understand the importance of clear and accurate scientific communication, so we thank the reviewer again for his thoroughness.

Comment #16: Line 384 I do not understand that sentence...."Elemental analysis of nitrogen and phosphorus (NValderrama and P Valderrama) was done with 0.05 g/mat following a high temperature persulfate oxidation [54]."

Response: We thank the reviewer for his observation. The sentence was revised and corrected (164-166, 455).

In previous line 384, we used to refer to the determination of nitrogen an

---

## [Editor Report · Decision Letter 1]

4 Dec 2024

What defines a photosynthetic microbial mat in western Antarctica?

PONE-D-24-02443R1

Dear Dr. Falcón,

We’re pleased to inform you that your manuscript has been judged scientifically suitable for publication and will be formally accepted for publication once it meets all outstanding technical requirements.

Kind regards,

Andrea Franzetti

Academic Editor

PLOS ONE
---

## [Editor Report · Acceptance letter]

PONE-D-24-02443R1

PLOS ONE

Dear Dr. Falcón,

I'm pleased to inform you that your manuscript has been deemed suitable for publication in PLOS ONE. Congratulations! Your manuscript is now being handed over to our production team.

Kind regards,

on behalf of

Dr. Andrea Franzetti

Academic Editor

PLOS ONE